# Family Resilience and Adolescent Mental Health during COVID-19: A Moderated Mediation Model

**DOI:** 10.3390/ijerph19084801

**Published:** 2022-04-15

**Authors:** Ran Zhuo, Yanhua Yu, Xiaoxue Shi

**Affiliations:** 1Department of Applied Psychology, School of Humanities, Jilin Agricultural University, Changchun 130118, China; zhuoran@jlau.edu.cn (R.Z.); shixiaoxuejlnydx@outlook.com (X.S.); 2Faculty of Education, Northeast Normal University, Changchun 130024, China; 3Center for Faculty Development, Jilin Agricultural University, Changchun 130118, China

**Keywords:** adolescent mental health, family resilience, pandemic stress perception, meta-mood, COVID-19 pandemic

## Abstract

Background: The COVID-19 pandemic has impacted and is still impacting people’s lives, including physical and mental health. Family plays an important role in adolescent mental health due to the long staying at home. Aims: This paper aimed to investigate the impact of family resilience on adolescent mental health during the COVID-19 pandemic, and the mediation role of pandemic stress perception and the moderation role of meta-mood. Methods: A total of 2691 Chinese adolescents were recruited using convenient sampling. Their mental health, family resilience, pandemic stress perception and meta-mood were surveyed. Multivariate statistics were used to analyze the data. Results: Our results showed that (1) about 36.7% adolescents in our sample have some mental health problems; (2) family resilience can positively predict adolescent mental health, whereas pandemic stress perception can negatively predict mental health; (3) pandemic stress perception mediates the relationship between family resilience and adolescent mental health; (4) meta-mood moderates the relationship between family resilience and pandemic perception, i.e., the first half of the mediation role. Conclusions: Our results indicate that one can either improve family resilience or improve adolescents’ meta-mood to relieve adolescents’ mental health problems.

## 1. Introduction

COVID-19, originating in December 2019, has spread all over the world and is still affecting people’s daily lives. On 31 January 2020, the World Health Organization designated COVID-19 Pandemic as an “International Public Health Emergency” (World Health Organization, Novel Coronavirus (2019-nCoV) Situation report-1. https://www.who.int/docs/default-source/coronaviruse/situation-reports/20200121-sitrep-1-2019-ncov.pdf?sfvrsn=20a99c10_ accessed on 6 June 2021). The COVID-19 pandemic has not only threatened people’s physical health, but also had impacts on people’s mental health [1,2]. Ref. [3] has shown that most people have experienced different degrees of stress, anxiety and depression during the pandemic, and these feelings persist for a period of time. Adolescence is an important period in individual’s development, during which adolescents are sensitive to appearance, academic performance, interpersonal relationship and many other aspects, and have increasingly perceived pressures [4]. When the pandemic started, it was Chinese New Year when all Chinese students were on winter vacation. Learning was switched to online in the following semester, which was never happened before. Similar actions were taken all over the world during lockdowns. This was undoubtedly an unprecedented challenge for adolescents and their families. A long time staying at home made family an important factor for adolescents’ mental health. Therefore, studying the relationship between family resilience and adolescent mental health during COVID-19 has important implications for both adolescents and their families.

### 1.1. Adolescents’ Stress Perception and Metal Health during COVID-19

Due to the COVID-19 pandemic, adolescents’ daily life and learning patterns have changed significantly. On the one hand, they have to face the risk of being infected at any time, and some people may suffer from separation from their family members or even death of their loved ones. Under these external stressors, they are more likely to receive great psychological pressure, causing serious mental health problems [5]. A study in China has shown that adolescents were at high risk for mental health during the pandemic [6]. Similarly, [7] reported that 85.7% of adolescents showed changes in emotional states, such as irritability, restlessness, tension during the COVID-19 outbreak, and the incidence of depressive symptoms ranged from 22.6–43.7%. Ref. [8] indicated that 58% of Lithuanian adolescents were of good mental health after the second lockdown, whereas 19% of adolescents had depression risk. On the other, adolescents who are high school students have to adapt to the changes in their learning styles, online courses and assessments, as well as the entrance examinations for higher schools in July (postponed for one month due to the pandemic). Studies also demonstrated that adolescents’ mental health such as anxiety, learning pressure, maladjustment and emotional state generally rose under governments’ pandemic prevention and control policies. If these stress and anxious emotions were not effectively alleviated, mental health problems would occur [9]. In addition, family conflicts increased due to the increasing time and the limited living spaces that adolescents shared with their parents following the policy of ‘staying at home and not going out’ during the pandemic. Ref. [10] showed that parents reported about 22.2% more conflicts with their children, and 28.3% of parents reported being stressed about relationship challenges with their partners due to the pandemic. Therefore, the stress that adolescents feel during the pandemic may come from pandemic risk, study pressure and family relationship, which all have impacts on their mental health.

### 1.2. Family Resilience and Adolescent Mental Health

Although perceived stresses can have negative effects on mental health [11], individuals could stay healthy and do well facing the pandemic or adversity, which is inseparable from the function of resilience [12,13]. This study discussed resilience in the context of family and analyzed its influence on adolescent mental health. First, family itself is an important place where children and adolescents’ socialization processes take place, and it is also a very important social ecological system that affects the psychological development of adolescents [14]. Second, ‘home quarantine’ was used by governments as a coping strategy at the beginning of the pandemic. Family environment where adolescents live and study, and the relationship between family members would undoubtedly affect adolescents’ external behaviors and emotions [15]. Moreover, many researchers have realized the importance of family resilience for family members to adapt and develop in adversity [16,17]. There are two types of definitions for family resilience. First, family resilience is defined as the ability and strength of a family as a whole to cope with stress and crisis [16,18,19]). Second, family resilience is seen as the dynamic process in which families constantly balance pressure, demands and resources to survive and adapt to the environment [16,17]. Whatever the definition is, family resilience has three characteristics. First, family is treated as a whole. Second, there must be some adversity that the family is involved in. Third, the final adaptation to the adversity is achieved through family members’ joint efforts [17,20]. Ref. [20] found that families with high resilience are more in control under stressful situations; ref. [21] posited that family resilience can directly predict adolescents’ depression, loneliness and happiness, which has a compensatory effect. Most studies on family resilience and mental health primarily focused on left-behind children, families with diseases or families with economic difficulties, demonstrating that family resilience can help left-behind children, families with patients and other families in adversity to actively adapt and achieve good development [22]. In addition, protective factors in family resilience, such as family belief and family support, can help families move forward in crisis and protect the survival and well-being of the whole family [23]. Therefore, family resilience may have a significant impact on adolescents’ mental health during the COVID-19 pandemic.

Meanwhile, high family resilience can reduce the adverse effects of stress through the perception of pressures [24]. According to the family resilience system theory, the three most important parts of family functioning are family belief systems, organization patterns and communication processes [25]. Family resilience is nurtured by shared beliefs that help members understand crisis situations, lead positive lives, and provide transcendent spiritual values and purpose [23]. Families can achieve a sense of cohesion by redefining a crisis as a common challenge that is understandable, manageable and meaningful [26]. Normalizing and contextualizing family members’ pain, and viewing stressful events as natural or understandable can weaken family members’ responses and reduce feelings of blame, shame, and guilt [23]. Identifying and affirming the strength of family in difficulties help to deal with the feelings of helplessness, failure and despair, so individuals can get good adaptation and development. Ref. [27] summarized the characteristics of children with high resilience, and pointed out that children with high resilience cannot do well without the support and encouragement of family members. Although the arrival of the pandemic could make adolescents feel pressured from various sources, it is possible for them to treat the pressure and crisis as understandable and meaningful challenges if they live in families with high intimacy, good communication, strong faith. Their negative perception of pressures may be weakened. Through the above argument, it can be conjectured that family resilience could not only affect adolescents’ mental health directly, but also indirectly by reducing their perceived pandemic stress.

### 1.3. The Role of Meta-Mood

Meta-mood is the ability to notice, distinguish, reflect and control ones’ emotions, which can help individuals to identify and adjust negative emotions when facing stressful events and deal with negative events in a positive way [28]. It can improve individuals’ anxiety, mental health level and obtain more happiness [29]. In addition, meta-mood also affects self-harmony, social adaptation and psychological resilience [30]. The Trait Meta Mood Scale (TMMS) measures meta-mood from three dimensions: attention to emotions, emotional clarity and emotion repair [31]. It is reported that individuals vary in their meta-mood [28]. Ref. [32] showed that meta-mood can predict perceived stress. Moreover, for those who pay a lot of attention to their emotions and have difficulties in emotion repair, it is very likely that they perceive more stresses during the pandemic, even if their family resilience is high [32]. On the contrary, if individuals do not consider mood as relevant to anything and can easily repair their emotions, they may perceive less stress even if the family resilience is not so high [32]. Therefore, it is reasonable to hypothesize that the meta-mood has moderating effect on the relationship between family resilience and pandemic stress perception.

### 1.4. The Present Study

This study will investigate the associations between family resilience, pandemic stress perception, meta-mood and adolescent mental health under COVID-19 Pandemic. Specifically, we will explore the mediating role of pandemic stress perception in the relationship between family resilience and adolescent mental health and the moderating role of meta-mood on the link between family resilience and pandemic stress perception. Based on the previous discussion, we proposed the following hypotheses:

**Hypothesis** **1.***Pandemic stress perception has a negative effect on adolescent mental health*.

**Hypothesis** **2.***Family resilience has a positive effect on adolescent mental health*.

**Hypothesis** **3.***Pandemic stress perception mediates the impact of family resilience on adolescent mental health*.

**Hypothesis** **4.***Meta-mood moderates the first half of the mediation role that Pandemic stress perception plays*.

Putting the above hypotheses together, we get our conceptual model as depicted in Figure 1.

## 2. Materials and Methods

### 2.1. Participants

The participants were recruited from Jilin Province, Yunnan Province and other provinces in China by convenient sampling via Chinese social media WeChat between 15 October 2020 and 2 December 2020. During that period, most places in China lifted the lockdown restrictions. A total of 2711 questionnaires were distributed to junior (year 7–year 9) and senior high school (year 10 to year 12) students. All the questionnaires were returned with a response rate of 100%. 2691 of them were valid. The demographic information of the participants is presented in Table 1 as follows: 1244 male students (46.2%) and 1447 female students (53.8%); 1668 junior high school students (62%) and 1023 senior high school students (38%); The participants were 11–18 years old with an average age of 16.66 ± 1.77. Among the 2691 participants, 58 were infected and recovered, accounting 2.2%.

### 2.2. Procedures

Participants were recruited by word of mouth on the Chinese social media application WeChat. Introduction about the research was given at the beginning of the questionnaire detailing the purpose of the research. In addition, the participants were briefed about the anonymous and voluntary nature of the participation, and they can withdraw at any time. The research was approved by the first author’s research review board against the university’s research ethics guideline.

### 2.3. Measures

Most measures have been tested in a Chinese context. The adolescent pandemic stress perception questionnaire was developed by the research team, and its reliability and validity were rigorously tested.

#### 2.3.1. Family Resilience

The family resilience questionnaire in China context was developed by [33] to evaluate the performance of a family as a whole under pressure, consisting of four dimensions: (1) Perseverance (6 items), reflecting a family’s courage, perseverance and positive efficacy in adversities. (2) Harmony (6 items), reflecting close and harmonious relationship between family members, i.e., whether they respect and care for each other. (3) Openness (4 items), reflecting whether the family has a good social relationship, that is, if it is positive and optimistic, and constantly seeks to learn and change. (4) Support (4 items), reflecting the cooperation and mutual help between family members when facing difficulties. The questionnaire was used on a five-point Likert scale from “1 = completely disagree” to “5 = completely agree.” The reliability and validity of the questionnaire have been tested using samples from Chinese contexts [33]. The Cronbach α coefficient of the overall questionnaire was 0.94, and the Cronbach α coefficients of each dimension were 0.81, 0.87, 0.78, 0.81, respectively, indicating that the questionnaire had a good reliability.

#### 2.3.2. Pandemic Stress Perception Questionnaire

The assessment tool of pandemic stress perception was developed by the research team. Firstly, narrative interviews about pandemic stress were conducted with 6 adolescents from different places in China, and the authors extracted stressors that adolescents reported through qualitative analysis. Then, the Pandemic Stress Perception Questionnaire for adolescents was developed based on the SARS stress perception questionnaire developed by [34]. The scale consists of 20 items, including three dimensions: Pandemic panic (3 items), study pressure (9 items), family pressure (8 items). A 5-point score was used, ranging from “0 = did not happen”, “1 = not very serious” to “4 = very serious”. And rigorous psychometric evaluations (including preliminary test, item analysis, item correlation analysis, exploratory factor analysis, retest, confirmatory factor analysis, reliability and validity analysis, etc.) were conducted, demonstrating a good reliability and validity. The structural model of the questionnaire reached acceptable level: χ^2^/DF = 24.05, RMSEA = 0.093, GFI = 0.0.866, CFI = 0.916. Mean score was used with high score meaning high pandemic stress perception by adolescents. The Cronbach α coefficient of the scale was 0.946, and the Cronbach α coefficients for each dimension were 0.793, 0.942 and 0.944, respectively.

#### 2.3.3. Meta-Mood

The Trait Meta-mood Scale (TMMS) was developed by [28] and revised by [35] in Chinese context. The revised TMMS was used in this research, which contains 26 items in total, including three dimensions, i.e., attention to feelings, emotional clarity and emotion repair. A five-point Likert scale was used ranging from “1 = Very inconsistent” to “5 = Very consistent” and the total score was added up for each dimension. The scale has a good reliability and validity. In this study, the Cronbach α coefficient of the scale was 0.893.

#### 2.3.4. Adolescent Mental Health

The General Health Questionnaire 12-item (GHQ-12) [36] was used. It consists of 12 items, half positive (e.g., “able to concentrate on whatever I do”) and half negative (e.g., “insomnia due to anxiety”), with a 4-point scale ranging from “1 = never” to “4 = often”. Reverse score was used for negative items. The total score ranges from 12 to 48, with a higher score indicating a better mental health, whereas an overall score below 33 is considered poor mental health [36]. In our sample, the Cronbach α coefficient is 0.765.

### 2.4. Data Analysis Methods

All the data were analyzed by SPSS 26 (IBM, New York, NY, USA). First, the data were screened to check the suitability for statistical analysis. Missing data were checked and normality check for variables were done. Then, descriptive analysis was conducted for the variables. And variance analysis was used to test the differences of variables against demographic variables. Next, the main effect was tested by using linear regression. The mediation effect and moderated mediation effect were tested using the macro “PROCESS” for SPSS.

## 3. Results

### 3.1. Data Screening

Prior to the formal analysis, data should be screened regarding to their suitability to statistical analysis. First, missing data was processed. In our sample, all the questionnaires were filled completely without missing data. Second, the normality of variables was checked. We calculated the skewness and kurtosis for family resilience (−1.492, 2.309), Pandemic stress perception (0.719, 0.11), meta-mood (−0.133, 2.533), mental health (0.14, −0.313). We also conducted the both the KS and Shapiro–Wilk normality tests for the four variables. Family resilience, pandemic stress perception, meta-mood and mental health were statistically significant (*p* < 0.001). So, we believe that the variables follow normal distribution [37]. In addition, Harman’s single factor test results showed that there were 11 factors with trait roots greater than 1. The first factor could explain 25.92% of the cumulative variation, which was less than recommended threshold of 40%, indicating that there was no serious common method bias [38]. After the data screening, we then proceeded with the statistical data analysis. In the following analysis, the mean values of the variables were used.

### 3.2. Descriptive Analysis

#### 3.2.1. Adolescent Mental Health

According to the General Health Questionnaire 12-item (GHQ-12) assessment standard (the score range is between 12 and 48, and the higher the score, the better the mental health), the score below 33 indicates poor mental health. Table 2 shows the distributions of health status across junior and senior high schools and gender. Data showed that the overall mental health of adolescents during the pandemic was good, with 36.7% of them having certain mental health problems. In addition, the distribution of mental health in junior high school students is significantly different from senior high school students, whereas the distribution of mental health in gender is not statistically significant.

#### 3.2.2. Adolescents’ Pandemic Stress Perception

As shown in Figure 2, the means and standard deviations of the three dimensions were as follows: Pandemic panic (2.18 ± 0.96), study pressure (2.46 ± 1.08), family pressure (1.54 ± 0.85). ANOVA test showed that the difference among the three dimensions was statistically significant (*p* < 0.00). Post hoc analysis showed that study pressure is the highest and family pressure is the lowest among the three dimensions.

#### 3.2.3. Family Resilience

The descriptive analysis of family resilience is presented in Table 3. It can be seen that the family resilience during the pandemic was relatively good, and the average score of each subscale was close to 4, indicating that the families of adolescents during the pandemic were resilient overall.

#### 3.2.4. Adolescents’ Meta-Mood

The meta-mood scores are summarized in Table 4. It can be seen that the meta-mood scores of adolescents during the pandemic period are at a middle and upper level. According to the score of each dimension, adolescents had slightly higher scores in emotional repair than in emotional clarity and in emotional attention.

### 3.3. Analysis of Differences in Adolescents’ Mental Health

ANOVA was used to analyze if there is significant difference of adolescents’ mental health in gender, grade, whether they were in graduating classes during the pandemic period. The results are shown in Table 5. It can be seen that there was no significant gender difference in adolescents’ mental health. The mental health of senior high school students was worse than that of junior high school students, and the mental health of graduating students was worse than that of non-graduating students. Overall, the mental health level of year 7 was the highest and year 10 was the lowest.

### 3.4. Analysis of Differences in Adolescents’ Stress Perception

It can be seen from Table 6 that there was no significant gender difference in adolescents’ stress perception. The stress perception of senior high school students was worse than that of junior high school students, and the stress perception of graduating students was worse than that of non-graduating students. On the whole, the stress perception of year 10 is highest and year 7 is lowest.

### 3.5. Correlation Analysis

Correlation analysis was conducted to study the correlations between variables which are presented in Table 7. It indicates that there was significant positive correlation between family resilience and adolescents’ mental health, meaning that the higher the family resilience is, the better the adolescent mental health. There was a significant negative correlation between pandemic stress perception and mental health. It suggests that the more stress adolescents felt, the lower their mental health level. In addition, meta-mood was positively related to family resilience and mental health, and negatively related to stress perception.

### 3.6. Main Effects

Linear regression was used with family resilience and pandemic stress perception as independent variables and adolescent mental health as dependent variables. Considering grade was significant in the ANOVA analysis, we put it into the model as a control variable. The regression results are shown in Table 8. It can be seen that family resilience had a significant positive effect on adolescents’ mental health (*β* = 0.185 *p* < 0.001), and Pandemic stress perception has a significant negative effect on adolescents’ mental health (*β* = −0.187 *p* < 0.001). Therefore, hypothesis 1 and hypothesis 2 are supported. Family resilience and Pandemic stress perception have significant effect on adolescent mental health.

### 3.7. Mediation Effect

We used model 4 of SPSS macro PROCESS to test the mediating effect of pandemic stress perception on family resilience and adolescent mental health [39]. The results showed that family resilience can significantly predict adolescents’ pandemic stress perception (*a* = −0.136, *SE* = 0.018, *p* < 0.001). Family resilience can significantly predict adolescent mental health (*c* = 0.185, *SE* = 0.009, *p* < 0.001. Pandemic stress perception can significantly predict adolescent mental health, (*b* = −0.187, *SE* = 0.01, *p* < 0.001). The model output is presented in Figure 3, which shows that the relationship between family resilience and adolescent metal health is mediated by pandemic stress perception.

The standardized indirect effect on this model is (−0.136) × (−0.186) = 0.025. We used bootstrapping procedure to test the significance of this indirect effect with 5000 bootstrapped samples and computed the 95% *CI* (confidence interval) by determining the indirect effects at the 2.5th and 97.5th percentiles as illustrated in Table 9.

It showed that pandemic stress perception had a significant mediating effect between family resilience and adolescents’ mental health, *ab* = 0.025 Boot SE = 0.004, 95% confidence interval [0.017, 0.034]. The proportion of mediating effect to total effect was *ab*/(*ab* + *c*) = 11.8%. Therefore, the Hypothesis 3 was supported. Pandemic stress perception mediates the relationship between family resilience and mental health.

### 3.8. Moderated Mediation Effect

We used model 7 of SPSS macro program PROCESS to test the moderating effect of meta-mood on the link between family resilience and pandemic stress perception. The results are presented in Table 10. Bootstrap analysis results showed that the indirect effect of the mediation test did not contain 0 (LLCI = 0.02 ULCI = 0.05), indicating that the mediation effect of the pandemic stress perception was significant, and the size of the mediation effect was 0.04. In addition, the direct effect of the independent variable family resilience on the mental health of adolescents was significant, and the interval (LLCI = 0.16, ULCI = 0.20) did not include 0. This suggests that pandemic stress perception plays a partially mediating role in the relationship between family resilience and adolescents’ mental health.

At the same time, the first half of the mediating effect of pandemic stress perception between family resilience and adolescent mental health was moderated by meta-mood. In order to clearly reveal the interaction effect of family resilience and meta-mood, simple slope analysis was estimated using the “pick a point” method [40]. We calculated the regression coefficients of family resilience on pandemic stress perception when the meta-mood was high (mean + 1 SD) and low (mean − 1 SD). For descriptive purposes, we plotted the predicted pandemic stress perception against family resilience separately for low and high levels of meta-mood (See Figure 4). Results showed that when meta-mood was low, family resilience could not significantly predict epidemic stress perception (*Bsimple* = 0.002, *SE* = 0.02, *p*
*>* 0.05); when meta-mood was high, the family resilience significantly predicted epidemic stress perception (*Bsimple* = −0.19, SE = 0.04, *p* < 0.001). Therefore, the interaction supports the reinforcement hypothesis of “protection factor-protection factor model”.

In sum, the mediation process of family resilience impacting adolescents’ mental health through pandemic stress perception was moderated by meta-mood. For adolescents with high meta-mood, the indirect effects of family resilience on adolescents’ mental health through pandemic stress perception was 0.035 which was significant, with BootSE = 0.008, and 95% confidence interval [0.02, 0.05]; for adolescents with low meta-mood, the indirect effect was −0.00 which was not significant, with BootSE = 0.004 and 95% confidence interval [−0.01, 0.00]. Therefore, Hypothesis 4 was supported.

## 4. Discussion

### 4.1. Adolescent Mental Health and Pandemic Stress Perception during the Pandemic

Our results showed that, although the overall adolescent mental health was good, a proportion (36.7%) of the samples got a low score, which means that the pandemic indeed had some impacts on adolescents’ lives. The main pressures were from study pressure and pandemic panic. This was because adolescents are in a key period of developing their outlooks on world, life and values, during which they have strong reliance on their parents and have weak adaptation capabilities. They cannot have a full picture of things and their self-judgement and adjustment capability are not strong enough to cope with the unprecedented events. When facing threats of the pandemic, their emotions and behaviors could be easily affected. This is in line with which showed that different degrees of emotional problems would emerge if adolescents stayed in a closed space for a long time. When adolescents have different opinions with their family members, conflicts could be easily occurred. Senior high school students had higher pressure than junior high school students and graduating students had higher pressure than non-graduating students in study pressure, pandemic panic and family pressure. This may be because higher-grade adolescents can use social media more frequently and can get more pandemic related information, which may increase their information anxiety [41]. Higher-grade students have more difficult assignments and study challenges which may increase their study pressure. In addition, most adolescents could not adapt to the online learning, resulting in low learning efficiency. This is particularly true for adolescents in graduating year and senior high school students with greater learning difficulties. However, the pressure level of senior high school students did not decrease with the increase in grade, the pressure level of freshmen in senior high school is the highest. This may be because the resilience and emotional adjustment of year 9 was worse than year 10 and year 11, so they felt more academic frustration when pandemic disrupted their preparation for entry examination for a higher school and home study cannot achieve the expected effects. Therefore, they were facing both pressures of pandemic and examination.

The proportion of unhealthy adolescents (36.7%) is different from existing research from Canada which identified that 67–70% of adolescents had mental health problems [1]. This could be caused by the different measurements used and the differences of the pandemic prevention policies between the two countries. From the variance analysis, we can see that mental health had no significant differences on gender, whereas it had significant differences on grades; in particular, senior high school students had worse mental health than junior high school students and graduating students had worse mental health than non-graduating students. This result was consistent with the case of adolescents’ pressures, which indicated that senior students were more prone to mental health problems.

In sum, besides the direct physical threats, the impact of COVID-19 pandemic on mental health cannot be ignored due to the various pressures including but not limited to the uncertainties of the pandemic, constantly delayed re-opening of schools, maladaptation to online learning, conflicts with family members. The results suggested that families and schools should understand adolescents’ emotions and stresses, help them to relieve the negative perception of pressures, increase their capability of decrease pressure perception to reduce the impact of the COVID-19 pandemic.

### 4.2. The Impact of Family Resilience and Pandemic Stress Perception on Mental Health

Our results have shown that family resilience can positively predict adolescent mental health and pandemic stress perception can negatively predict adolescent mental health. This is consistent with the literature [24]. Facing the threats to adolescents’ mental health brought by the panic and pressures from the pandemic, families would schedule their resources to make adjustment. However, not all families can come out of the pressures easily, and family resilience plays an important role in this process. This could be explained from the following aspects. First, the belief system of family resilience can make adolescents see the pandemic positively, and see the disadvantages as temporary, changeable, so they can deal with the negative impacts by the pandemic positively, adapt to the new environment, enabling them to progress towards a better and positive direction [22]. Another research showed that positive people can solve their current problem in a better way, reducing pressures and overcoming the difficulties, which is an important factor predicting their positive development [42]. Second, family organizing system and communication system can make family members get along well with each other, making them feel happy and comfortable. Adolescents can actively seek help and get more support from family, which make them show stronger psychological resilience when facing risks [43]. Meanwhile, literature has shown that more stresses could lead to mental health problems such as depression [44]. This is the same during the pandemic. Although the stressors are different, the perceived stress during the pandemic can significantly impact adolescents’ mental health as shown in our result.

### 4.3. The Mediation Role of Pandemic Stress Perception

Research has shown that stressful life event is a risk factor to adolescents’ mental health [5,45]. Our research supported the existing research and showed that family resilience cannot only impact adolescents’ mental health directly, but also through reducing adolescents’ perceived stresses indirectly.

Specifically, our research found that family resilience had negative prediction on adolescents’ pandemic stress perception. The resiliency model of family stress indicates that members in a family under pressure (big risk event, or daily hassles etc.) can unite together through holding various social psychological resources to cope with risks so that family functions well, which in turn makes family members to adjust and adapt from the imbalanced status caused by the risk to a balanced status. Adolescents from families with better family resilience could feel less panic of the pandemic (e.g., worse off of the pandemic), less study pressure (e.g., disruption of the preparation for examination), less family pressure (e.g., increased conflicts between members) [46]. On the one hand, family resilience can help adolescents see the uncertainties and damages positively and reduce their negative emotions. On the other hand, family conflicts, which are usually high when members are staying in a closed space for a long time, are less in families with better resilience [10,47]. This could be due to their higher intimacy and better communications, which help adolescents control their emotions and feel less pressures from family, but rather more support and sense of belonging from family. A study on family resilience and development of high school students after a trauma from Sichuan Wenchuan earthquake disaster area showed that students from better family resilience can see the disaster of the earthquake in a positive way, and can feel positive power from their family so as to find positive ways to cope with the disaster [48].

### 4.4. The Moderation Role of Meta-Mood

Our result showed that meta-mood can moderate the impact of family resilience on pandemic stress perception. The moderation role supported the reinforcement hypothesis in the “protection factor- protection factor model” [49]. This means that meta-mood can increase the protection of family resilience on pandemic stress perception, supporting the stress buffer model, i.e., meta-mood as a protection factor for individuals coping with pressures could buffer or reduce its bad affects [50]. Literature has investigated the interaction model between meta-mood and negative life events (such as pandemic pressure) [51] which showed that negative life events’ negative prediction on happiness and mental health depended on individual’s meta-mood level.

On the one hand, staying home during the pandemic increased adolescents’ usage of social media, which increased the possibility of information anxiety. Adolescents’ negative cognition on pandemic information is one of the important sources of anxiety. Meta-mood is the ability of an individual recognizing and controlling his/her emotion [52]. Meta-mood can help adolescents’ get more positive mental capital (confidence, hope, positive, resilience), enabling them to actively deal with and resolve the impact of a negative event [29,53]. Furthermore, adolescents with higher meta-mood usually pay less attention to their emotions, know their emotion status clearly and are able to repair their bad emotion [52]. This will strengthen their relationship with family members. Therefore, when they face pressure events, they can feel positive power from family and get more social supports [54]. Therefore, compared with adolescents with lower meta-mood, those with higher meta-mood can more easily have positive cognition on the pandemic and see the pandemic as a temporary and surmountable challenge. Family intimacy, cohesiveness and adaptability would be higher. When facing pressures and conflicts, family members can actively communicate, understand each other, support each other to find a solution. This cannot only make adolescents have more resources to cope with difficulties, but also promote family’s recovery capabilities from disaster.

### 4.5. Implications and Suggestions

Our results have important implications to the prevention and avoidance of adolescents’ mental health problems during the pandemic. First, adolescents’ mental health is impacted by their stresses in their ecosystem. With the increase in perceived stresses, their mental health gets worse. Therefore, it is necessary to consider adolescents’ stresses from family, school and society systematically and comprehensively to conduct intervention on their crisis in time. Second, our results showed that family resilience and meta-mood are internal mechanisms for adolescents’ mental health, which suggests parents should create a good family environment, using democratic parenting and increase positive emotional accompany. Meanwhile, parents should care about adolescents’ emotion control capability, making them realize that they are the master of their emotions. Schools should strengthen the training of applying emotion adjustment policies in pressured contexts (for example, mindfulness training, behavioral therapy) and reduce their negative emotional experience.

## 5. Conclusions

Using a sample from Chinese adolescents, this paper investigated the relationship between family resilience and adolescent mental health. We found that (1) there was a proportion of adolescents who had mental health problems, but the overall situation was good; (2) after controlling grades, we found that pandemic stress perception can significantly negatively predict adolescents’ mental health, whereas family resilience can positively predict mental health. (3) Adolescents’ stress perception mediates the impact of family resilience on adolescents’ mental health. (4) The impact of family resilience on pandemic stress perception was moderated by meta-mood.

This research is not without limitations. First, we used self-reported measurements to evaluate adolescents’ mental health, which may not be able to measure the true degree of mental health such as anxiety or depression. Therefore, mental health experts should be invited to evaluate adolescents’ mental health. Second, adolescents’ stresses during the pandemic and mental health are changing processes during the development of the pandemic. Therefore, longitudinal research methods should be used to trace adolescents’ stresses and mental health at different time points during the pandemic to explore the relationship between the variables. Third, our samples were drawn using convenient sampling method, which may not be representative. Therefore, other sampling methods should be used to select various samples.

## Figures and Tables

**Figure 1 ijerph-19-04801-f001:**
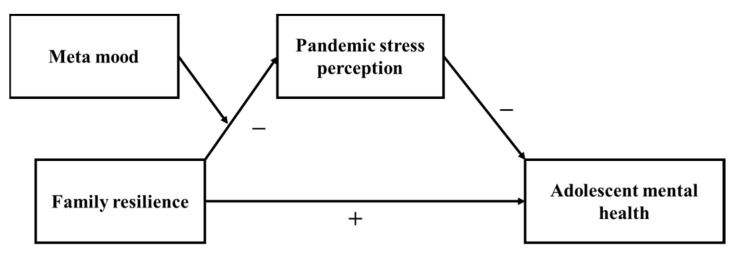
Conceptual model.

**Figure 2 ijerph-19-04801-f002:**
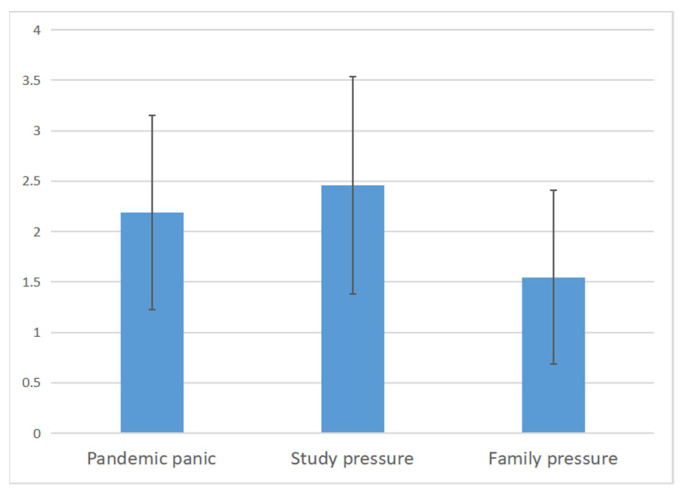
Adolescents’ Pandemic stress perception.

**Figure 3 ijerph-19-04801-f003:**
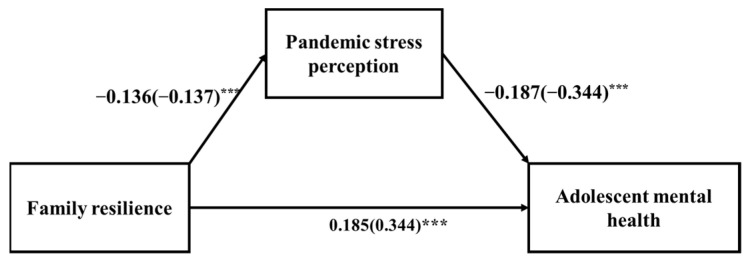
Standardized regression coefficients for the relationship between family resilience and adolescent mental health mediated by pandemic stress perception. *** *p* < 0.001.

**Figure 4 ijerph-19-04801-f004:**
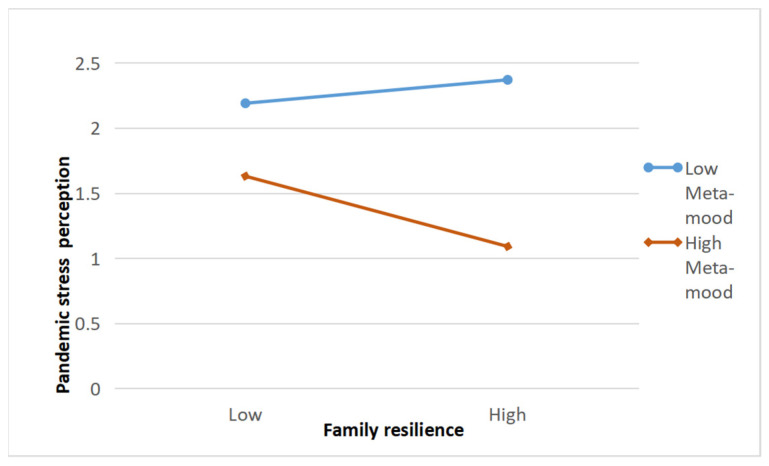
The moderating effect of meta-mood on the relationship between family resilience and pandemic stress perception.

**Table 1 ijerph-19-04801-t001:** Demographic information (*N* = 2691).

Variable	Value	Number	Percentage
Gender	Boy	1244	46.2%
	Girl	1447	53.8%
Grade	Year 7	765	28.4%
	Year 8	534	19.8%
	Year 9	369	13.7%
	Year 10	69	2.6%
	Year 11	81	3.0%
	Year 12	873	32.5%
Infection condition	Uninfected	2633	97.8%
	Infected	58	2.2%

**Table 2 ijerph-19-04801-t002:** Adolescents’ mental health during the pandemic.

		Good Mental Health	Poor Mental Health	χ^2^	*p*
Junior	Number	1139	529	93.34	0.00
	Percentage	68.3%	31.7%		
Senior	Number	564	459		
	Percentage	55.1%	44.9%		
Boy	Number	768	476	2.38	0.12
	Percentage	61.7%	38.3%		
Girl	Number	935	512		
	Percentage	64.6%	35.4%		
Total	Number	1703	988		
	Percentage	63.3%	36.7%		

**Table 3 ijerph-19-04801-t003:** Family resilience situation.

	*N*	Minimum Value	Maximum Value	*M*	*SE*
Tenacity	2691	1.00	5.00	4.26	0.90
Harmony	2691	1.00	5.00	4.35	0.90
Openness	2691	1.00	5.00	3.90	0.94
Support	2691	1.00	5.00	4.38	0.89
Overall	2691	1.00	5.00	4.23	0.82

**Table 4 ijerph-19-04801-t004:** Adolescents’ meta-mood.

	*N*	Minimum Value	Maximum Value	*M*	*SE*
Emotion attention	2691	1.90	5.00	3.49	0.54
Emotion recognition	2691	1.70	5.00	3.48	0.58
Emotion recovery	2691	1.00	5.00	3.90	0.76
Overall score	2691	1.92	5.00	3.58	0.52

**Table 5 ijerph-19-04801-t005:** ANOVA analysis of mental health.

		*N*	*M*	*SD*	*t/F*	*p*
Gender	Boy	1244	2.89	0.43	0.006	0.995
	Girl	1447	2.89	0.45		
School level	Junior high	1668	2.80	0.44	8.73 ***	0.000
	Senior high	1023	2.95	0.43		
Graduating (non-graduating)	Non-graduating	1449	2.93	0.43	−4.53 ***	0.000
	Graduating	1242	2.86	0.45		
Grade	Year 7	765	2.99	0.18	37.49 ***	0.000
	Year 8	534	2.92	0.43		
	Year 9	369	2.90	0.47		
	Year 10	69	2.55	0.28		
	Year 11	81	2.67	0.37		
	Year 12	873	2.83	0.43		

*** *p* < 0.001.

**Table 6 ijerph-19-04801-t006:** ANOVA analysis of stress perception.

		*N*	*M*	*SD*	*t/F*	*p*
Gender	Boy	1244	2.07	0.83	0.56	0.57
	Girl	1447	2.05	0.79		
School level	Junior high	1668	1.84	0.74	−0.11 ***	0.000
	Senior high	1023	2.59	0.98		
Graduating (non-graduating)	Non-graduating	1449	2.21	0.78	9.28 ***	0.000
	Graduating	1242	1.93	0.81		
Grade	Year 7	765	1.82	0.75	75.11 ***	0.000
	Year 8	534	1.89	0.76		
	Year 9	369	1.82	0.71		
	Year 10	69	2.99	0.94		
	Year 11	81	2.27	0.92		
	Year 12	873	2.38	0.75		

*** *p* < 0.001.

**Table 7 ijerph-19-04801-t007:** Correlation analysis among variables (*N* = 2691).

	*M*	*SD*	1	2	3	4
Family resilience	4.22	0.82	1			
Pandemic stress perception	2.06	0.81	−0.19 **	1		
Meta-mood	3.50	0.520	0.58 **	−0.28 **	1	
Adolescent mental health	2.89	0.44	0.41 **	−0.41 **	0.61 **	1

** *p* < 0.01.

**Table 8 ijerph-19-04801-t008:** Regression analysis (*N* = 2691).

Value	*R* ^2^	*B (SE)*	*β*	*p*
Constant	0.29	2.52(0.04)		0.00
Year 8		−0.04(0.02)	−0.04 **	0.02
Year 9		−0.07(0.02)	−0.05 **	0.003
Year 10		−0.06(0.04)	−0.02	0.201
Year 11		−0.12(0.04)	−0.04 **	0.006
Year 12		−0.01(0.01)	−0.01	0.535
Family resilience		0.18(0.00)	0.34 ***	0.000
Pandemic stress perception		−0.18(0.01)	−0.34 ***	0.000

** *p* < 0.01, *** *p* < 0.001.

**Table 9 ijerph-19-04801-t009:** Effects in the mediation model.

	Effect	*SE*	*p*	95% *CI*s	
				Lower *CI*	Upper *CI*
Total effect	0.211	0.01	0.000	0.192	0.229
Direct effect	0.185	0.01	0.000	0.167	0.203
Indirect effect	0.025	0.004		0.017	0.034
Standardized indirect effect	0.047	0.008		0.032	0.062

Note: The bootstrap sample size is 5000.

**Table 10 ijerph-19-04801-t010:** The moderated mediation effect model.

Regression Equation		Fitting Index		Regression Coefficient		
Result Variable	Predictive Variable	*R*²	*F*	*β*	95%*CI*	*t*
Pandemic Stress perception	Year 8	0.18	78.19	0.053	[−0.02, 0.13]	1.27
	Year 9			−0.04	[−0.13, 0.04]	−0.92
	Year 10			0.98 ***	[0.79, 1.16]	10.44
	Year 11			0.29 ***	[0.12, 0.46]	3.36
	Year 12			0.48 ***	[0.41, 0.55]	13.18
	Family resilience			−0.09 ***	[−0.14, −0.04]	−3.43
	Meta mood			−0.28 ***	[−0.35, −0.21]	−8.12
	Family resilience × Meta-mood			−0.18 ***	[−0.25, −0.11]	−4.98
Adolescent mental health	Year 8	0.29	157.91	−0.04 ***	[−0.08, −0.00]	−2.29
	Year 9			−0.07 ***	[−0.11, −0.02]	−3.00
	Year 10			−0.06	[−0.15, 0.03]	−1.27
	Year 11			−0.12 ***	[−0.20, −0.03]	−2.72
	Year 12			−0.01	[−0.049, 0.02]	−0.62
	Family resilience			0.18 ***	[0.16, 0.20]	20.45
	Pandemic Stress perceived			−0.18 ***	[−0.20, −0.16]	−19.52

Note: *N* = 2691. Bootstrap sample size: 5000. *** *p* < 0.001.

## Data Availability

The dataset in this research can be obtained upon request.

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
