# Peer review of "Family Resilience and Adolescent Mental Health during COVID-19: A Moderated Mediation Model"

_ijerph, 2022, doi:10.3390/ijerph19084801_

Round 1
Reviewer 1 Report
Convenient sampling - the authors probably meant convenience sampling
Where does the number of 2711 participants come from?
Did the authors count the minimal sample size?
I think the characteristics of the included population should be broader described. There could be more factors affecting analysed domains that need to be controlled, e.g. social status or history of mental disorders.
How the reliability and validity of the adolescent pandemic stress perception questionnaire were tested?
In my opinion, dividing people into healthy and unhealthy based only on the questionnaire is not reasonable and stigmatising.
Regarding Table 2 - why did not the authors provide any statistical test for the difference?
Regarding Fig. 2, what is the statistical test result for differences?
It will be more readable to have the description of statistical methods in the method section.
It is unclear where this formula comes from and what c means. Is it 0.185 from the previous page?
p10 l348 It is unclear where this conclusion 'Therefore, the Hypothesis 3 was supported' comes from.
Reviewer 2 Report
Attached is a document with my comments and suggestions. I was very interested in reading the article and of course the practical implications of its findings are very important, especially considering the variables analyzed: mental health of adolescents during the pandemic, family, perception of stress, etc.

Reviewer 3 Report
In this paper, the authors investigate how family resilience impacts adolescent mental health during the COVID-19 pandemic and discuss the mediation role of pandemic stress perception and the moderation role of meta-mood. The authors hypothesize a causal model and collect data with questionnaires. The hypotheses are tested with linear regression utilizing the collected data.
In general, the manuscript is well structured and not difficult to follow. The authors introduce sufficient background information related to adolescents' stress perception, family resilience, mental health, and meta-mood. The description of the methods and results is clear. The authors also discussed the main findings adequately.
It looks like the authors do not sufficiently introduce the rationale for why some of the hypotheses are introduced, and I am not confident whether some of the hypotheses are plausible.
The authors hypothesize that pandemic stress perception mediates the impact of family resilience on adolescent mental health. However, it looks like it might be more reasonable to hypothesize that family resilience is the mediator between pandemic stress perception and adolescent mental health. Adolescents perceive stress, which potentially impacts their mental health. Familly with high resilience might reduce such impact as a mediator. The authors need to better explain why hypothesis 3 is more reasonable than the alternative hypothesis.
In hypothesis 4, the authors hypothesize that meta-mood moderates family resilience's impact on pandemic stress perception. However, it looks like meta-mood might be a moderator between stress perception and mental health. Meta-mood is related to the ability to adjust negative emotions when facing stressful events. It looks more reasonable to assume adolescents have perceived stress; however, those adolescents with better meta-mood can adjust negative emotions associated with the stress. As a consequence, stress has less impact on their mental health. The authors might also need to clarify why hypothesis 4 is more appropriate.
Round 2
Reviewer 1 Report
The manuscript has been revised according to the comments.
This manuscript is a resubmission of an earlier submission. The following is a list of the peer review reports and author responses from that submission.